# KERNEL GRAPH CONVOLUTIONAL NEURAL NETS

## ABSTRACT

Graph kernels have been successfully applied to many graph classification problems. Typically, a kernel is first designed, and then an SVM classifier is trained based on the features defined implicitly by this kernel. This two-stage approach decouples data representation from learning, which is suboptimal. On the other hand, Convolutional Neural Networks (CNNs) have the capability to learn their own features directly from the raw data during training. Unfortunately, they cannot handle irregular data such as graphs. We address this challenge by using graph kernels to embed meaningful local neighborhoods of the graphs in a continuous vector space. A set of filters is then convolved with these patches, pooled, and the output is then passed to a feedforward network. With limited parameter tuning, our approach outperforms strong baselines on 7 out of 10 benchmark datasets, and reaches comparable performance elsewhere. Code and data are publicly available[1].

## 1 INTRODUCTION

Graphs are powerful structures that can be used to model almost any kind of data. Social networks (Kitsak et al., 2010), textual documents (Mihalcea & Tarau, 2004), the World Wide Web (Page et al., 1999), chemical compounds (Mahé & Vert, 2009), and protein-protein interaction networks (Borgwardt et al., 2007), are all examples of data that are commonly represented as graphs. As such, graph classification is a very important task, with numerous significant real-world applications. However, due to the absence of a unified, standard vector representation of graphs, graph classification cannot be tackled with classical machine learning algorithms.

Kernel methods offer a solution to those cases where instances cannot be readily vectorized. The trick is to define a suitable object-object similarity function (known as a kernel function). Then, the matrix of pairwise similarities can be passed to a kernel-based supervised algorithm such as the Support Vector Machine (Cortes & Vapnik, 1995) to perform classification. With properly crafted kernels, this two-step approach was shown to give state-of-the-art results on many datasets (Shervashidze et al., 2011), and has become standard and widely used. One major limitation of the graph kernel + SVM approach, though, is that representation and learning are two *independent* steps. In other words, the features are precomputed in separation from the training phase, and are not optimized for the downstream task.

Conversely, Convolutional Neural Networks (CNNs) learn their own features from the raw data during training, to maximize performance on the task at hand. CNNs thus provide a very attractive alternative to the aforementioned two-step approach. However, CNNs are designed to work on regular grids, and thus cannot process graphs.

We propose to address this challenge by extracting patches from each input graph via community detection, and by embedding these patches with graph kernels. The patch vectors are then convolved with the filters of a 1D CNN and pooling is applied. Finally, to perform graph classification, a fully-connected layer with a softmax completes the architecture.

We compare our proposed method with state-of-the-art graph kernels and a recently introduced neural architecture on 10 bioinformatics and social network datasets. Results show that our Kernel CNN model is very competitive, and offers in many cases significant accuracy gains.

---

[1] https://goo.gl/WG7nkD

## 2 RELATED WORK

### 2.1 GRAPH KERNELS

A kernel implicitly represents objects as vectors in a Hilbert space $\mathcal{H}$ (Schölkopf & Smola, 2001). Given a set $\mathcal{X}$, let $k : \mathcal{X}^2 \to \mathbb{R}$ be a positive semidefinite kernel function. Then, there exists a Hilbert space $\mathcal{H}$, known as the Reproducing Kernel Hilbert Space (RKHS) associated with $k$. Furthermore, there exists a mapping $\phi : \mathcal{X} \to \mathcal{H}$, such that, for any $(x, x') \in \mathcal{X}^2$ the kernel value $k(x, x')$ is equal to the inner product $\langle \phi(x), \phi(x') \rangle_{\mathcal{H}}$.

A graph kernel is a kernel defined on pairs of graphs. Graph kernels can be viewed as graph similarity functions, and currently serve as the dominant tool for graph classification. Most graph kernels compute the similarity between two networks by comparing their substructures, which can be specific subgraphs (Shervashidze et al., 2009), random walks (Kashima et al., 2003; Gärtner et al., 2003; Mahé et al., 2004; Borgwardt et al., 2005; Vishwanathan et al., 2010; Sugiyama & Borgwardt, 2015), cycles (Horváth et al., 2004), or paths (Borgwardt & Kriegel, 2005; Feragen et al., 2013), among others. The Weisfeiler-Lehman framework operates on top of existing kernels and improves their performance by using a relabeling procedure based on the Weisfeiler-Lehman test of isomorphism (Shervashidze et al., 2011). Recently, two other frameworks were presented for deriving variants of popular graph kernels (Yanardag & Vishwanathan, 2015a;b). Inspired by recent advances in NLP, they offer a way to take into account substructure similarity. Some graph kernels not restricted to comparing substructures of graphs but that also capture their global properties have also been proposed. Examples include graph kernels based on the Lovsz number and the corresponding orthonormal representation (Johansson et al., 2014), the pyramid match graph kernel that embeds vertices in a feature space and computes an approximate correspondence between them (Nikolentzos et al., 2017), and the Multiscale Laplacian graph kernel, which captures similarity at different granularity levels by considering a hierarchy of nested subgraphs (Kondor & Pan, 2016).

### 2.2 GRAPH CNNS

Extending CNNs to graphs has experienced a surge of interest in recent years, following the impressive results reached by CNNs in computer vision (Vinyals et al., 2015; Krizhevsky et al., 2012) and NLP (Kim, 2014). With the exception of (Tixier et al., 2017), in which graphs are represented as images using stacked bivariate histograms of their node embeddings and passed to a classical 2D CNN, most of the CNN-based graph classification approaches proposed in the literature introduce either operational or architectural modifications to CNNs.

A first class of methods use spectral properties of graphs. An early generalization of the convolution operator to graphs was based on the eigenvectors of the Laplacian matrix (Bruna et al., 2014). This work was later generalized to high-dimensional datasets, and to settings where the graph structure is not known *a priori* (Henaff et al., 2015). A more efficient model using Chebyshev polynomials approximation to represent the spectral filters was later presented (Defferrard et al., 2016). All of these methods, however, assume a fixed graph structure and are thus not applicable to our setting. The model of (Defferrard et al., 2016) was then simplified by using a first-order approximation of the spectral filters (Kipf & Welling, 2017), but within the context of a *node* classification problem (which again, differs from our *graph* classification setting).

Unlike spectral methods, spatial methods (Niepert et al., 2016; Vialatte et al., 2016) operate directly on the topology of the graph. The work closest to ours is probably (Niepert et al., 2016). To extract a set of patches from the input graph, the authors (1) construct an ordered sequence of vertices from the graph, (2) create a neighborhood graph of constant size for each selected vertex, and (3) generate a vector representation (patch) for each neighborhood using graph labeling procedures such that nodes with similar structural roles in the neighborhood graph are positioned similarly in the vector space. The extracted patches are then fed to a 1D CNN.

In contrast to the above work, we extract neighborhoods of varying sizes from the graph in a more direct and natural way (via community detection), and use graph kernels to normalize our patches. We present our approach in more details in the next section.

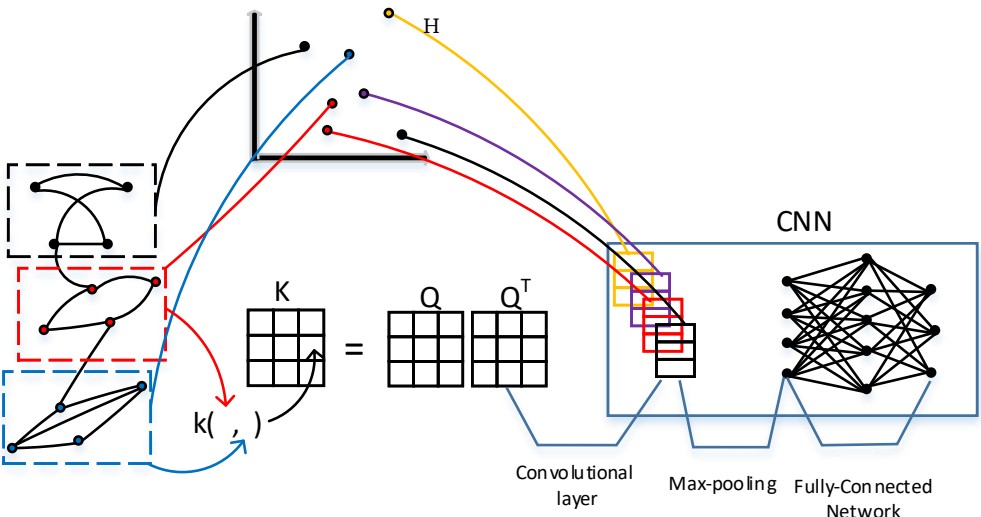

Figure 1: Overview of our Kernel Graph CNN approach.

# 3 PROPOSED APPROACH

In what follows, we present the main ideas and building blocks of our model. The overarching process flow is illustrated in Figure 1.

## 3.1 PATCH EXTRACTION AND NORMALIZATION

Many types of real world data are regular grids, and can thus be decomposed into units that are inherently ordered along spatial dimensions. This makes the task of patch extraction easy, and normalization unnecessary. For example, in computer vision (2D), meaningful patches are given by instantiating a rectangle window over the image. Furthermore, for all images, pixels are uniquely ordered along width and height, so there is a correspondence between the pixels in each patch, given by the spatial coordinates of the pixels. This removes the need for normalization. Likewise, in NLP, words in sentences are uniquely ordered from left to right, and a 1D window applied over the text provides again natural regions.

However, graphs do not exhibit such an underlying grid-like structure. They are irregular objects for which there exist no canonical ordering of the elementary units (nodes). Hence, generating patches from graphs, and normalizing them so that they are comparable and combinable, is a very challenging problem. To address these challenges, our approach leverages *community detection* and *graph kernels*.

**Patch extraction with community detection**. There is a large variety of approaches for sampling from graphs. We can extract subgraphs for all vertices (which may be computationally intractable for large graphs) or for only a subset of them, such as the most central ones according to some metric. Furthermore, subgraphs may contain only the hop-1 neighborhood of a root vertex, or vertices that are further away from it. They may also be walks passing through the root vertex. A more natural way is to capitalize on *community detection* algorithms (Fortunato & Hric, 2016), as the clusters correspond to meaningful graph partitions. Indeed, a community typically corresponds to a set of vertices that highly interact among each other, as expressed by the number and weight of the edges between them, compared to the other vertices in the graph. In this paper, we employ the Louvain clustering algorithm, which extracts non-overlapping communities of various sizes from a given graph (Blondel et al., 2008). This multilevel algorithm aggregates each node with one of its neighbors such that the gain in modularity is maximized. Then, the groupings obtained at the first step are turned into nodes, yielding a new graph. The process iterates until a peak in modularity is attained and no more change occurs. Note that since our goal here is only to sample relevant local

neighborhoods from the graph, we could have used any other state-of-the-art community detection algorithm. We opted for Louvain as it is very fast and scalable.

**Patch normalization with graph kernels**. After extracting the subgraphs (communities) from a given input graph, standardization is necessary before being able to pass them to a CNN. We can define this step as that of *patch normalization*. To this purpose, we leverage graph kernels, as described next. Note that since the steps below do not depend on the way the subgraphs were obtained, we use the term *subgraph* (or *patch*) rather than *community* in what follows, to highlight the generality of our approach.

Let $\mathcal{G} = \{G_1, G_2, \ldots, G_N\}$ be the collection of input graphs. Let $\mathcal{S}_1, \mathcal{S}_2, \ldots, \mathcal{S}_N$ be the sets of subgraphs extracted from graphs $G_1, G_2, \ldots, G_N$ respectively. Since the number of subgraphs extracted from each graph may depend on the graph (like in our case with the Louvain community detection algorithm), these sets are likely to vary in size (see Table 2).

Given a graph $G_i$ and its set of subgraphs $\mathcal{S}_i$, we denote by $S_i^j$ the $j^{th}$ subgraph and by $P_i$ the total number of subgraphs extracted from $G_i$. Let then $\mathcal{S} = \{S_i^j : i \in \{1, 2, \ldots, N\}, j \in \{1, 2, \ldots, P_i\}\}$ be the set of subgraphs extracted from all the graphs in the collection, and $P$ its cardinality. Let finally $K \in \mathbb{R}^{P \times P}$ be the symmetric positive semidefinite kernel matrix constructed from $\mathcal{S}$ using a graph kernel $k$. Since the total number $P$ of subgraphs for all the graphs in the collection is very large, populating the full kernel matrix $K$ and factorizing it to obtain low-dimensional representations of the subgraphs is $\mathcal{O}(P^3)$. Fortunately, the Nyström method (Williams & Seeger, 2000) allows us to obtain $Q \in \mathbb{R}^{P \times p}$ (with $p \ll P$) such that $K \approx QQ^\top$ at the reduced cost of $\mathcal{O}(p^2 P)$, by using only a small subset of $p$ columns (or rows) of the kernel matrix. The rows of $Q$ are low-dimensional representations of the subgraphs and serve as our normalized patches.

## 3.2 GRAPH PROCESSING

**Convolution**. To process a given input graph, numerous filters are convolved with the normalized representations of the patches contained in the graph. For example, for a given filter $w \in \mathbb{R}^p$, a feature $c_i$ is generated from the $j^{th}$ patch of graph $G_i$ $z_i^j$ as:

$$c_j = \sigma(w^\top z_i^j)$$

where $\sigma$ is an activation function. In this study, we used the identity function $\sigma(c) = c$, as we observed no difference in results compared to nonlinear activations. Therefore, when applied to a patch $z_i^j$, the pooling operation corresponds to the inner product $\langle w, z_i^j \rangle$. We show in the Appendix that the filters live in the RKHS of the kernel $k$ that was used to normalize the patches.

By convolving $w$ with all the normalized patches of the graph, the following feature map is produced:

$$c = [c_1, c_2, \ldots, c_{P_{max}}]^\top$$

where $P_{max} = \max(P_i : i \in \{1, 2, \ldots, N\})$ is the largest number of subgraphs extracted from a given graph in the collection. For graphs featuring less than $P_{max}$ patches, zero-padding is employed.

Note that this approach is similar to concatenating all the vector representations of the patches contained in a given graph (padding where necessary), thus obtaining a single vector representation of the graph, and sliding a unidimensional filter of same size as the length of a single patch vector, with stride equal to the size of the filter (i.e., no overspanning patches).

**Pooling**. We then apply a max-pooling operation over the feature map, thus retaining only the maximum value of $c$, $\max(c_1, c_2, \ldots, c_{P_{max}})$, as the signal associated with $w$. The intuition is that some subgraphs of a graph are good indicators of the class the graph belongs to, and that this information will be picked up by the max-pooling operation.

## 3.3 PROCESSING NEW GRAPHS

When provided with a never-seen graph (at test time), we first sample subgraphs from it (here, via community detection), and then project them to the feature space of the subgraphs in the training set. Given a new subgraph $S^j$, its projection is computed as follows:

$$z^j = Q^\dagger v$$

where $Q^\dagger \in \mathbb{R}^{p \times P}$ is the pseudoinverse of $Q \in \mathbb{R}^{P \times p}$ and $v \in \mathbb{R}^P$ is the vector containing the kernel value between $S^j$ and all subgraphs in the training set (those contained in set $\mathcal{S}$). The dimensionality of the emerging vector ($p$) is the same as that of the normalized patches in the training set. Thus, this vector can be convolved with the filters of the CNN as previously described.

## 3.4 CHANNELS

Rather than selecting one graph kernel in particular to normalize the patches, several kernels can be jointly used. The different representations provided by each kernel can then be passed to the CNN through different channels, or *depth* dimensions. Intuitively, this can be very beneficial, as each kernel might capture different, complementary aspects of similarity between subgraphs. We experimented with the following popular kernels:

• **Shortest path kernel (SP)** (Borgwardt & Kriegel, 2005): to compute the similarity between two graphs, this kernel counts how many pairs of shortest paths have the same source and sink labels, and identical length, in the two graphs. The runtime complexity for a pair of graphs featuring $n_1$ and $n_2$ nodes is $\mathcal{O}(n_1^2 n_2^2)$.
• **Weisfeiler-Lehman subtree kernel (WL)** (Shervashidze et al., 2011): for a certain number $h$ of iterations, this kernel performs an exact matching between the compressed multiset labels of the two graphs, while at each iteration it updates these labels. It requires $\mathcal{O}(hm)$ time for a pair of graphs with $m$ edges. For our experiments, we set the $h$ parameter of the WL kernel equal to 5.

This gave us two single channel models (KCNN SP, KCNN WL), and one model with two channels (KCNN SP+WL).

## 4 EXPERIMENTAL SETUP

### 4.1 SYNTHETIC DATASET

**Dataset**. As previously mentioned, the intuition is that our proposed KCNN model is particularly well suited for settings where some regions in the graphs are good class indicators. To empirically verify this claim, we created a dataset featuring 1000 synthetic graphs generated as follows:
First, we generate an Erdos-Rényi graph (Erdös & Rényi, 1959) with number of vertices sampled from $[100, 200] \cap \mathbb{Z}$ with uniform probability, and edge probability equal to 0.1. We then add to the graph either a 10-clique or a 10-star graph by connecting their vertices with probability 0.1. An $n$-clique is a graph that contains $n$ vertices, and every pair of vertices is connected by an edge. An $n$-star graph is a tree consisting of $n$ vertices where one vertex has degree $n - 1$ and all the other vertices have degree 1. The first class of the dataset is finally made of the graphs containing a 10-clique, while the second class features the graphs containing a 10-star subgraph. The two classes are of equal size (500 graphs each).

**Baselines**. We compared our model against the shortest-path kernel (SP) (Borgwardt & Kriegel, 2005), the Weisfeiler-Lehman subtree kernel (WL) (Shervashidze et al., 2011), and the graphlet kernel (GR) (Shervashidze et al., 2009). The first two kernels were presented in subsection 3.4. The graphlet kernel counts identical pairs of subgraphs up to a certain size $k$ in the two graphs. Exhaustive enumeration of these subgraphs requires $\mathcal{O}(n^k)$ time. For large graphs, enumerating all graphlets is infeasible and sampling schemes are usually employed.

**Configuration**. We performed 10-fold cross-validation. The $C$ parameter of the SVM (for all graph kernel baselines) and the number of iterations (for the WL kernel baseline) were optimized on a 90-10 split of the training set of each fold. We respectively chose from $\{10^n; n \in \text{seq}(-3, 4, \text{by} = 1)\}$ and $\{3, 4, 5, 6\}$. For the graphlet kernel, we sampled 1000 graphlets of size up to 6 from each graph. For our proposed KCNN, we used an architecture with one convolutional-pooling block followed by a fully connected layer with 128 units. The `ReLU` activation was used, and regularization was ensured with dropout (0.5 rate) (Srivastava et al., 2014). A final `softmax` layer was added to complete the architecture. It outputs a probability distribution over classes. The dimensionality of the normalized patches (number of columns in $Q$) was set to $p = 100$, and we used 256 filters. The model was implemented in Python 3.6 using the PyTorch[2] library. Batch size was set to 64, and

---

[2] https://github.com/pytorch/pytorch

Table 1: Classification accuracy of state-of-the-art graph kernels: shortest path (SP), graphlet (GR), and Weisfeiler-Lehman subtree (WL); and the single and multichannel variants of our approach (KCNN), on the synthetic dataset.

| Method / Dataset | SP | GR | WL | KCNN SP | KCNN WL | KCNN SP+WL |
|---|---|---|---|---|---|---|
| SYNTHETIC | 75.47 | 69.34 | 65.88 | 98.20 | 97.25 | **98.40** |

Table 2: Summary of the 10 real-world datasets used in our experiments.

| | ENZYMES | NCI1 | PROTEINS | PTC-MR | D&D | IMDB BINARY | IMDB MULTI | REDDIT BINARY | REDDIT MULTI-5K | COLLAB |
|---|---|---|---|---|---|---|---|---|---|---|
| Max # vertices | 126 | 111 | 620 | 109 | 5748 | 136 | 89 | 3782 | 3648 | 492 |
| Min # vertices | 2 | 3 | 4 | 2 | 30 | 12 | 7 | 6 | 22 | 32 |
| Avg. # vertices | 32.63 | 29.87 | 39.05 | 25.56 | 284.32 | 19.77 | 13.00 | 429.61 | 508.50 | 74.49 |
| Max # edges | 149 | 119 | 1049 | 108 | 14267 | 1249 | 1467 | 4071 | 4783 | 40119 |
| Min # edges | 1 | 2 | 5 | 1 | 63 | 26 | 12 | 4 | 21 | 60 |
| Avg. # edges | 62.14 | 32.30 | 72.81 | 25.96 | 715.66 | 96.53 | 65.93 | 497.75 | 594.87 | 2457.34 |
| Total # communities | 2856 | 20168 | 5197 | 1193 | 14981 | 2752 | 2591 | 44015 | 114152 | 15499 |
| Avg. # communities | 4.76 | 4.90 | 4.66 | 3.46 | 12.71 | 2.75 | 1.72 | 22.00 | 22.83 | 3.09 |
| Avg. # vertices per community | 6.85 | 6.08 | 8.36 | 4.11 | 22.35 | 7.18 | 7.52 | 19.52 | 22.26 | 24.03 |
| # labels | 3 | 37 | 3 | 19 | 82 | - | - | - | - | - |
| # graphs | 600 | 4110 | 1113 | 344 | 1178 | 1000 | 1500 | 2000 | 4999 | 5000 |
| # classes | 6 | 2 | 2 | 2 | 2 | 2 | 3 | 2 | 5 | 3 |

the number of epochs and learning rate were optimized by performing 10-fold cross-validation on the training set of each fold. Number of epochs was chosen from the set $\{25, 50, 100, 200\}$, and the learning rate from $\{10^{-3}, 10^{-4}\}$. All experiments were run on a single machine consisting of a 3.4 GHz Intel Core i7 CPU with 16 GB of RAM and an NVidia GeForce Titan Xp GPU.

**Results**. We report in Table 1 average prediction accuracies of our three models in comparison to the baselines. The results shown in Table 1 confirm our hypothesis that our proposed model (KCNN) can identify those areas in the graphs that are most predictive of the class labels, as its three variants achieved accuracies greater than $98\%$. Conversely, the baseline kernels failed to discriminate between the two categories. Hence, it is clear that in such settings, our model is more effective than existing methods. Finally, it is interesting to note that the performance of KCNN with 16 filters was very close to that reported (with 256 filters).

## 4.2 REAL-WORLD DATASETS

We also evaluated the performance of our approach on five bioinformatics and five social network datasets, publicly available at (Kersting et al., 2016). Notice that the bioinformatics datasets are labeled (labels on vertices), while the social interaction datasets are not. Table 2 shows summary statistics about these datasets.

**Bioinformatics datasets**. ENZYMES (Borgwardt et al., 2005) is a dataset of 600 protein tertiary structures obtained from the BRENDA enzyme database. Each enzyme is a member of one of the Enzyme Commission top level enzyme classes (EC classes) and the task is to correctly assign the enzymes to their classes. NCI1 (Wale et al., 2008) contains 4110 chemical compounds screened for activity against non-small cell lung cancer and ovarian cancer cell lines. PROTEINS (Borgwardt et al., 2005) consists of 1113 proteins represented as graphs where vertices are secondary structure elements and there is an edge between two vertices if they are neighbors in the amino-acid sequence or in 3D space. PTC-MR (Toivonen et al., 2003) is a dataset of 344 organic molecules where classes indicate carcinogenicity for male rats. D&D (Dobson & Doig, 2003) consists of 1178 protein structures classified into enzymes and non-enzymes.

**Social network datasets**. All these datasets were put together by (Yanardag & Vishwanathan, 2015a). IMDB-BINARY and IMDB-MULTI are movie collaboration datasets. The vertices of each graph represent actors and two vertices are linked by an edge if the corresponding actors appear

Table 3: 10-fold CV average test set classification accuracy ($\pm$ standard deviation) of state-of-the-art graph kernels: shortest path (SP), graphlet (GR), random walk (RW), and Weisfeiler-Lehman subtree (WL); Deep Kernels (Yanardag & Vishwanathan, 2015a) (unless specified, with the best performing kernel); a state-of-the-art graph CNN (PSCN $k$=10) (Niepert et al., 2016); and the single and multichannel variants of our approach (KCNN); on 10 bioinformatics (top) and social network (bottom) datasets. Best performance per dataset in **bold**, among the variants of our Kernel CNN method underlined.

| Dataset / Method | ENZYMES | NCI1 | PROTEINS | PTC-MR | D&D |
|---|---|---|---|---|---|
| SP | 40.10 ($\pm$ 1.50) | 73.00 ($\pm$ 0.51) | 75.07 ($\pm$ 0.54) | 58.24 ($\pm$ 2.44) | > 3 days |
| GR | 26.61 ($\pm$ 0.99) | 62.28 ($\pm$ 0.29) | 71.67 ($\pm$ 0.55) | 57.26 ($\pm$ 1.41) | 78.45 ($\pm$ 0.26) |
| RW | 24.16 ($\pm$ 1.64) | > 3 days | 74.22 ($\pm$ 0.42) | 57.85 ($\pm$ 1.30) | > 3 days |
| WL | 53.15 ($\pm$ 1.14) | 80.13 ($\pm$ 0.50) | 72.92 ($\pm$ 0.56) | 56.97 ($\pm$ 2.01) | 77.95 ($\pm$ 0.70) |
| Deep Kernels | **53.43** ($\pm$ 0.91) | **80.31** ($\pm$ 0.46) | 75.68 ($\pm$ 0.54) | 60.08 ($\pm$ 2.55) | NA |
| PSCN $k = 10$ | NA | 76.34 ($\pm$ 1.68) | 75.00 ($\pm$ 2.51) | 62.29 ($\pm$ 5.68) | 76.27 ($\pm$ 2.64) |
| KCNN SP | 46.35 ($\pm$ 0.39) | 75.70 ($\pm$ 0.31) | 74.27 ($\pm$ 0.22) | **62.94** ($\pm$ 1.69) | 76.63 ($\pm$ 0.09) |
| KCNN WL | 43.08 ($\pm$ 0.68) | 75.83 ($\pm$ 0.25) | **75.76** ($\pm$ 0.28) | 61.52 ($\pm$ 1.41) | 75.80 ($\pm$ 0.07) |
| KCNN SP + WL | 48.12 ($\pm$ 0.23) | 77.21 ($\pm$ 0.22) | 73.79 ($\pm$ 0.29) | 62.05 ($\pm$ 1.41) | **78.83** ($\pm$ 0.29) |

| Dataset / Method | IMDB BINARY | IMDB MULTI | REDDIT BINARY | REDDIT MULTI-5K | COLLAB |
|---|---|---|---|---|---|
| GR | 65.87 ($\pm$ 0.98) | 43.89 ($\pm$ 0.38) | 77.34 ($\pm$ 0.18) | 41.01 ($\pm$ 0.17) | 72.84 ($\pm$ 0.28) |
| Deep GR | 66.96 ($\pm$ 0.56) | 44.55 ($\pm$ 0.52) | 78.04 ($\pm$ 0.39) | 41.27 ($\pm$ 0.18) | 73.09 ($\pm$ 0.25) |
| PSCN $k = 10$ | 71.00 ($\pm$ 2.29) | 45.23 ($\pm$ 2.84) | **86.30** ($\pm$ 1.58) | 49.10 ($\pm$ 0.70) | 72.60 ($\pm$ 2.15) |
| KCNN SP | 69.60 ($\pm$ 0.44) | 45.99 ($\pm$ 0.23) | 77.23 ($\pm$ 0.15) | 44.86 ($\pm$ 0.24) | 70.78 ($\pm$ 0.12) |
| KCNN WL | 70.46 ($\pm$ 0.45) | 46.44 ($\pm$ 0.24) | 81.85 ($\pm$ 0.12) | **50.04** ($\pm$ 0.19) | **74.93** ($\pm$ 0.14) |
| KCNN SP + WL | **71.45** ($\pm$ 0.15) | **47.46** ($\pm$ 0.21) | 78.35 ($\pm$ 0.11) | 44.63 ($\pm$ 0.18) | 74.12 ($\pm$ 0.17) |

in the same movie. Each graph is the ego-network of an actor, and the task is to predict which genre an ego-network belongs to. REDDIT-BINARY and REDDIT-MULTI-5K are social interaction datasets, where graphs represent online discussion threads crawled from Reddit. Each vertex corresponds to a user, and two users are connected with an edge if one of them responded to at least one comment from the other. The task is to classify graphs into either communities or subreddits. Finally, COLLAB is a scientific collaboration dataset containing ego-networks of different researchers from three subfields of Physics, and the task is to determine the subfield to which the collaboration graph of each researcher belongs.

**Baselines**. We evaluated our model in comparison with the shortest-path kernel (SP) (Borgwardt & Kriegel, 2005), the random walk kernel (RW) (Gärtner et al., 2003), the graphlet kernel (GR) (Shervashidze et al., 2009), and the Weisfeiler-Lehman subtree kernel (WL) (Shervashidze et al., 2011). The SP, WL and GR kernels were presented in subsections 3.4 and 4.1. Given a pair of graphs, the RW kernel performs random walks on both graphs, and counts the number of matching walks. The computational complexity of the RW kernel is $\mathcal{O}(n^3)$ (Vishwanathan et al., 2010). Since the experimental setup is the same, we also report the results of (Yanardag & Vishwanathan, 2015a) (Deep Graph Kernels) and (Niepert et al., 2016) (Graph CNN, PSCN $k = 10$), for comparison purposes.

**Configuration**. Same as 4.1 above.

**Results**. The 10-fold cross-validation average test set accuracy of our approach and the baselines is reported in Table 3. Our approach outperforms all baselines on 7 out of the 10 datasets. In some cases, the gains in accuracy over the best performing competitors are considerable. For instance, on the IMDB-MULTI, COLLAB, and D&D datasets, we offer respective absolute improvements of 2.23, 2.33, and 2.56 in accuracy over the best competitor, the state-of-the-art graph CNN (PSCN $k = 10$). Finally, it should be noted that on the IMDB-MULTI dataset, every variant of our architecture outperforms all the baselines.

Table 4: 10-fold cross validation runtime of our approaches on the 10 real-world graph classification datasets.

|  | ENZYMES | NCI1 | PROTEINS | PTC-MR | D&D | IMDB BINARY | IMDB MULTI | REDDIT BINARY | REDDIT MULTI-5K | COLLAB |
|---|---|---|---|---|---|---|---|---|---|---|
| KCNN SP | 28" | 4' 26" | 42" | 22" | 54" | 36" | 1' 41" | 5' 29" | 15' 2" | 7' 2" |
| KCNN WL | 53" | 4' 54" | 48" | 22" | 1' 33" | 41" | 58" | 5' 22" | 14' 23" | 8' 58" |
| KCNN SP+WL | 1' 13" | 5' 1" | 53" | 25" | 1' 46" | 45" | 1' 44" | 9' 57" | 24' 28" | 10' 24" |

Overall, our Kernel CNN model reaches better performance than the classical graph kernels (SP, GR, RW, and WL), showing that the ability of CNNs to learn their own features during training is superior to disjoint feature computation and learning. It is true that our approach also comprises two disjoint steps. However, the first step is only a *data preprocessing* step, where we extract neighborhoods from the graphs, and normalize them with graph kernels. The features used for classification are then learned *during training* by our neural architecture, unlike the GK + SVM approach, where the features, given by the kernel matrix, are computed in advance, independently from the downstream task.

**Single-channel variants**. Our two single-channel architectures performed comparably on the bioinformatics datasets, while the KCNN WL variant was superior on the social network datasets. On the REDDIT-BINARY, REDDIT-MULTI-5K and COLLAB datasets, KCNN WL also outperforms the multichannel architecture, with quite wide margins.

**Multi-channel variants**. The multi-channel architecture (KCNN SP + WL) leads to better results on 5 out of the 10 datasets, showing that capturing subgraph similarity from a variety of angles sometimes helps.

**Runtimes**. We also report the time cost of our three models in Table 4. Runtime includes all steps of the process: patch extraction, path normalization, and 10-fold cross validation procedure. We can see that the computational complexity of the proposed models is not high. Our most computationally intensive model (KCNN SP+WL) took less than 25 minutes to perform the full 10-fold cross validation procedure on the largest dataset (REDDIT-MULTI-5K). Moreover, in most cases, the running times are lower or comparable to the ones of the state-of-the-art Graph CNN and Deep Graph Kernels models (Niepert et al., 2016; Yanardag & Vishwanathan, 2015a).

## 5 CONCLUSION AND NEXT STEPS

In this paper, we proposed a deep learning framework for graph classification with convolutional neural networks, Kernel Graph CNN. We extract meaningful patches from the input graphs with community detection, and use graph kernels to normalize the patches. This addresses the disjoint representation/learning limitation of the traditional graph kernel + SVM approach, while jointly capitalizing on the flexibility of graph kernels to process irregular objects, and on the unrivaled representational power and learning ability of CNNs. Furthermore, we show that patches can be normalized with a combination of multiple graph kernels to increase expressiveness. Results show that our model outperforms state-of-the-art baselines on 7 datasets out of 10. Finally, our framework offers the advantage of being very general: any graph kernel can be used, many different graph kernels can be combined, and the subgraphs can be obtained via any graph sampling method (not necessarily via community detection).

Ideally, we would like filters to be represented as graphs. Since convolution is just a dot product between a patch and a filter, computing the convolution would then simply require evaluating the kernel function on the two graphs. To generate those filters, we could use an algorithm that extracts frequent subgraphs from a collection of graphs, or we could sample a set of subgraphs from the extracted communities. However, backpropagating the error during training to those graph filters, and updating them accordingly, is a major challenge. We would also like to investigate the behavior of other methods for extracting patches, such as for instance overlapping community detection algorithms. Another important problem that needs to be solved is how to arrange patches in an order such that they can be convolved with overlapping filters (currently, convolution is performed independently for each filter). Finally, we would like to experiment with other neural network architectures, such as recurrent ones.

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

## 6 APPENDIX

### 6.1 FILTERS LIVE IN RKHS

We will show next that any filter $w$ with $||w|| < \infty$ learned by our network belongs to the RKHS of the employed graph kernel $k$.

Given two subgraphs $S_i^j$ and $S_{i'}^{j'}$ extracted from $G_i$ and $G_i'$ and their associated normalized patches $z_i^j$ and $z_{i'}^{j'}$, it holds that

$$\langle z_i^j, z_{i'}^{j'} \rangle = k(S_i^j, S_{i'}^{j'}) = \langle \phi(S_i^j), \phi(S_{i'}^{j'}) \rangle_{\mathcal{H}}$$

Let $\mathcal{Z} = \{z_i^j : i \in \{1, 2, \ldots, N\}, j \in \{1, 2, \ldots, P_i\}\}$ be the set containing all patches of the input graphs. Then, $\mathrm{Span}(\mathcal{Z})$ is either the space of all vectors in $\mathbb{R}^P$ if the rank of the kernel matrix is $P$ or the space of all vectors in $\mathbb{R}^P$ whose last $t$ components are zero if the rank of the kernel matrix is $P - t$ where $t > 0$. Then, given a patch $z_i^j$, vector $w$ is contained in the span of the set $\mathcal{Z}$, hence

$$\sigma(w^\top z_i^j) = \langle w, z_i^j \rangle = \langle \sum_{i'=1}^{N} \sum_{j'=1}^{P_i} a_{i'}^{j'} z_{i'}^{j'}, z_i^j \rangle = \sum_{i'=1}^{N} \sum_{j'=1}^{P_i} a_{i'}^{j'} \langle z_{i'}^{j'}, z_i^j \rangle = \sum_{i'=1}^{N} \sum_{j'=1}^{P_i} a_{i'}^{j'} k(S_{i'}^{j'}, S_i^j)$$

which shows that the filters live in the RKHS associated to graph kernel $k$. For other smooth activation functions, one can also show that the filters will be contained in the corresponding RKHS of the kernel function (Zhang et al., 2017).

