# OpenReview forum: "Kernel Graph Convolutional Neural Nets"
_ICLR.cc/2018/Conference — Reject_

### Official Review · AnonReviewer2 · 2017-11-26
**Clearly written but the evaluation is not thorough**

**Rating:** 5
**Confidence:** 5

**Review:**

This paper proposes a graph classification method by integrating three techniques, community detection, graph kernels, and CNNs.

* This paper is clearly written and easy to follow. Thus the clarity is high.

* The originality is not high as the application of neural networks for graph classification has already been studied elsewhere and the proposed method is a direct combination of three existing methods, community detection, graph kernels, and CNNs.

* The quality and the significance of this paper it not high due to the following reasons:
- The motivation is misleading in two folds.
  First, the authors say that the graph kernel + SVM approach has a drawback due to two independent processes of graph representation and learning.
  However, the parameters included in respective graph kernel is usually optimized via the SVM classification, hence they are not independent with each other.
  Second, the authors say that the proposed method addresses the above issue of independence between graph representation and learning.
  However, it also uses the two-step procedure as it first obtain the kernel matrix K via graph kernels and then apply CNN for classification, which is fundamentally the same as the existing approach.
  Although community detection is used before graph kernels, such subgraph extraction process is already implicitly employed in various graph kernels.
  I recommend to revise and clarify this point.
- In experimental evaluation, why several kernels including SP, RW, and WL are not used in the latter five datasets?
  This missing experiment significantly deteriorate the quality of empirical evaluation and I strongly recommend to add results for such kernels.
- It is mentioned that the parameter h is fixed to 5 in the WL kernel. However, it is known that the performance of the WL kernel depends on the parameter and it should be tuned by cross-validation.
  In contrast, parameters (number of epochs and the learning rate) are tuned in the proposed method. Thus the current comparison is not fair.
- In addition to the above point, how are parameters for GR and RW?
- Runtime is shown in Table 4 but there is no comparison with other methods. Although it is mentioned in the main text that the proposed method is faster than Graph CNN and Depp Graph Kernels, there is no concrete values and this statement is questionable (Runtime will easily vary due to the hardware configuration).

* Additional comment:
- Why is the community detection step needed? What will happen if K is directly constructed from given N graphs and what is the advantage of using not the original graphs but extracted subgraphs?
- In the first step of finding characteristic subgraphs, frequent subgraph mining can be used instead community detection.
  Frequent subgraph mining is extensively used in various methods for classification of graph-structured data, for example:
  * Tsuda, K., Entire regularization paths for graph data, ICML 2007.
  * Thoma, M. et al., Discriminative frequent subgraph mining with optimality guarantees, Statistical Analysis and Data Mining, 2010
  * Takigawa, I., Mamitsuka, H., Generalized Sparse Learning of Linear Models Over the Complete Subgraph Feature Set, IEEE Transactions on Pattern Analysis and Machine Intelligence, 2017
  What is the advantage of using the community detection compared to frequent subgraph mining or other subgraph enumeration methods?

---

### Official Review · AnonReviewer1 · 2017-11-27
**A well written, clear paper presening a method of using convolution neural networks for classifying arbitrary graphs.**

**Rating:** 5
**Confidence:** 4

**Review:**

The paper presents a method of using convolution neural networks for classifying arbitrary graphs. The authors proposed the following methodology
1) Extract subgraph communities from the graphs, known as patches
2) For each patch generate a graph kernel representation and subsampled them using nystrom method, producing the normalized patches
3) Passes the set of normalized patches as input to the CNN

The paper is well written, proposes an interesting and original idea, provides experiments with real graph datasets from two domains, bioinformatics and social sciences, and a comparison with SoA algorithms both graph kernels and other deep learning architectures. Although the proposed algorithm seems to outperform on 7 out of 10 datasets, the performances are really close to the best SoA algorithm. Is there any statistical significance over the gain in the performances? It's not   really clear from the reported numbers. Moreover, the method makes an strong assumption that the graph is strongly characterized by one of its patches, ie its subgraph communities, which might not be the case in arbitrary graph structures, thus limiting their method. I am not really convince about the preprocessing step of patch extraction. Have the authors  tried to test what is the performance of graph kernel representation in the complete graph as input to the CNN, instead of a set of patches? Moreover, although the authors claim that typical graph kernel methods are two-stage approached decoupling representation from learning, their proposal also folds into that respect, as representation is achieved in the preprocessing step of patching extractions and normalization, while learning is achieved by the CNN. Finally, it is not also clear to me the what are the communities reported in Table 2 for the  bioinformatics datasets. Where they come from and what do they represent?

---

### Official Review · AnonReviewer3 · 2017-11-27
**Minor technical contribution to representation learning, evaluation not convincing**

**Rating:** 4
**Confidence:** 5

**Review:**

The authors propose a method for graph classification by combining graph kernels and CNNs. In a first step patches are extracted via community detection algorithms.  These are then transformed into vector representation using graph kernels and fed to a neural network. Multiple graph kernels may serve as different channels. The approach is evaluated on synthetic and real-world graphs.

The article is well-written and easily comprehensible, but suffers from several weak points:

* Features are not learned directly from the graphs, but the approach merely weights graph kernel features.
* The weights refer to the RKHS and filters are not easily interpretable.
* The approach is similar in spirit to Niepert, Ahmed, Kutzkov, ICML 2016 and thus incremental.
* The experiments are not convincing: The improvement over the existing work is small on real-world data sets. The synthetic classification task essentially is to distinguish a clique from star graph and not very meaningful. Moreover, a comparison to at least one of the recent approaches similar to "Convolutional Networks on Graphs for Learning Molecular Fingerprints" (Duvenaud et al., NIPS 2015) or "Message Passing Neural Networks" (Gilmer et al., 2017)  would be desirable.

Therefore, I cannot recommend the paper for acceptance.

---

### Decision · Program_Chairs · 2018-01-29
**ICLR 2018 Conference Acceptance Decision**

**Decision:**

Reject

**Comment:**

The reviewers were unanimous in their assessment that the paper was not ready for publication in ICLR.  Their concerns included:
 - lack of novelty over Niepert, Ahmed, Kutzkov, ICML 2016
 - The approach learns combinations of graph kernels and its expressive capacity is thus limited
 - The results are close to the state of the art and it is not clear whether any improvement is statistically significant.

The authors have not provided a response to these concerns.